# The Role of Cardiac Resynchronization Therapy for the Management of Functional Mitral Regurgitation

**DOI:** 10.3390/cells11152407

**Published:** 2022-08-04

**Authors:** Eleonora Russo, Giulio Russo, Mauro Cassese, Maurizio Braccio, Massimo Carella, Paolo Compagnucci, Antonio Dello Russo, Michela Casella

**Affiliations:** 1Department of Cardiovascular Disease, IRCCS Casa Sollievo della Sofferenza, 71013 San Giovanni Rotondo, Italy; 2Department of Biomedicine and Prevention, Policlinico Tor Vergata, University of Rome, 00133 Rome, Italy; 3Department of Cardiac Surgery, Casa Sollievo della Sofferenza, 71013 San Giovanni Rotondo, Italy; 4Scientific Research Department, IRCCS Casa Sollievo della Sofferenza, 71013 San Giovanni Rotondo, Italy; 5Cardiology and Arrhythmology Clinic, University Hospital “Ospedali Riuniti Umberto I-Lancisi-Salesi”, 60126 Ancona, Italy; 6Department of Biomedical Sciences and Public Health, University Hospital ”Umberto I-Lancisi-Salesi”, Marche Polytechnic University, 60126 Ancona, Italy; 7Department of Clinical, Special and Dental Sciences, University Hospital “Umberto I-Lancisi-Salesi”, Marche Polytechnic University, 60126 Ancona, Italy

**Keywords:** cardiac resynchronization therapy, functional mitral regurgitation, dyssynchrony, heart failure, LV remodeling

## Abstract

Valve leaflets and chordae structurally normal characterize functional mitral regurgitation (FMR), which in heart failure (HF) setting results from an imbalance between closing and tethering forces secondary to alterations in the left ventricle (LV) and left atrium geometry. In this context, FMR impacts the quality of life and increases mortality. Despite multiple medical and surgical attempts to treat FMR, to date, there is no univocal treatment for many patients. The pathophysiology of FMR is highly complex and involves several underlying mechanisms. Left ventricle dyssynchrony may contribute to FMR onset and worsening and represents an important target for FMR management. In this article, we discuss the mechanisms of FMR and review the potential therapeutic role of CRT, providing a comprehensive review of the available data coming from clinical studies and trials.

## 1. Introduction

Functional mitral regurgitation (FMR) is a pathological condition in which the mitral valve is structurally normal, and the disease results from valve deformation caused by left ventricular (LV) dysfunction and remodeling [1]. It can occur in both ischemic and non-ischemic heart disease [2,3]. The prevalence of ischemic FMR varies from 20 to 50% [4,5], and the presence of moderate or severe FMR is associated with increased morbidity and mortality, with a 3-fold increased risk of heart failure (HF) and a 1.6-fold increased risk of death at the 5 year follow-up [3]. On the other hand, non-ischemic FMR is observed in 50% of cases in patients with LV dysfunction (from 56 to 65%) and is associated with excess cardiac mortality. HF episodes with 3-and 2-fold increased risks, respectively [6,7].

FMR results from multiple factors: decreased contractility, ventricular remodeling, impairment of mitral annular function, and ventricular dyssynchrony. Additionally, FMR tends to progress over time due to chronic volume overload, according to the classical motto “mitral regurgitation begets mitral regurgitation” [8]. The vicious circle between cardiac remodeling and FMR may be potentially interrupted by either reverse LV or annular remodeling or a decrease in mitral regurgitation (MR) severity. Pharmacotherapy can influence both targets by reducing transmitral pressure gradients, preload, and afterload. However, once HF and MR become refractory to pharmacotherapy, the prognosis becomes poor [9,10], and, so far, no univocal and definitive treatment has been found to alter the disease’s trajectory [11]. Patients with HF and FMR eligible for cardiac resynchronization therapy (CRT) could benefit from resynchronization in terms of reduction of the degree of FMR [12,13], which could translate into increased survival and improved symptoms. However, LV dyssynchrony is only one component of a more complex disease involving many heterogeneous mechanisms. Moreover, it has been observed that a number of patients are poor responders to CRT and have no FMR improvement. In such cases, additional device treatments (e.g., percutaneous mitral regurgitation intervention) should be considered. 

The present review will focus on the clinical impact of CRT on FMR, examining data regarding post-procedure and long-term outcomes of CRT in patients with FMR. Furthermore, we aim to provide the rationale for using mitral percutaneous interventions in patients in which FMR improvement is not achieved with CRT. 

## 2. The Complex Nature of Functional Mitral Regurgitation

### 2.1. Imbalance between Tethering and Closing Forces

The main determinants of FMR are the complex and heterogeneous changes in the MV apparatus that may occur in HF. Normally, the MV leaflets close in a coaptation point within the annular plane and have a considerable overlap when they face one to the other (coaptation reserve). In FMR, the coaptation reserve is progressively reduced until coaptation is no more possible. The main determinant of the reduction of coaptation reserve is the predominance of tethering forces: LV remodeling and dilation cause papillary muscles (PMs) dislocation away from the mitral annulus [14]. Consequently, PMs tend to pull MV leaflets apically and/or posteriorly through the attached chordae tendinae, thus augmenting tethering forces (Figure 1). At the same time, closing forces are decreased, mainly due to LV systolic dysfunction and, in particular, LV dilatation causing incomplete MV closure [15].

This imbalance between the two opposing forces generates a typical phasic intra-beat variation in the time course of the regurgitant orifice area in FMR, known as the ‘loitering pattern.’ The orifice area and regurgitation are greater in early and late systole and lower in the mid systole when the higher peak LV pressure facilitates valve closure.

### 2.2. The Dynamic Nature of Functional Mitral Regurgitation

Beyond intra-beat variability, FMR severity may also have a beat-to-beat variation. This characteristic depends on the phasic changes in the balance between tethering and closing forces and on the physiologic and/or pharmacologic factors able to modify this equilibrium. Exercise contributes to a greater cardiac load and may also trigger dynamic geometric changes in the LV and mitral valve apparatus, thus causing MR worsening [16] (Figure 1). In particular, volume loading in isotonic exercise results in exercise-induced ventricular dilation (end-diastolic volume and end-systolic volume both tend to increase), whereas ejection fraction remains unchanged [17]. In addition, exercise may result in dyssynchronous contraction of the LV with (rate-dependent) conduction delay [18]. However, what contributes most to MR worsening is the exercise-induced increase of LV sphericity, leading to a greater PMs distance and consequently to mitral valve geometry modification [16]. In ischemic cardiomyopathy, the localization, and the extent of myocardial scar play an important role in dynamic MR changes. In an anterior wall myocardial infarction extending to the apical segments of the anterior wall, exercise induces a displacement of PMs apically, thus increasing coaptation depth and worsening MR. In patients with inferior wall infarction, exercise aggravates MR in a differently: regional wall motion abnormalities tether the mitral valve more posteriorly, increasing the annular dimension [17]. The echocardiographic quantification of MR during exercise may unmask greater severity and provide prognostic information by disclosing its dynamic characteristics. Indeed, larger increases in the degree of MR during exercise are associated with an increased risk of death and hospital admissions for worsening HF [19]. Pharmacologic factors also can influence MR changes. Indeed diuretics, by decreasing preload, lead to a reduction in ventricular size with a decrease in tethering forces and consequently in MR. Inotropic agents (i.e., dobutamine infusion) modify dP/dt and the closing forces, thus reducing MR.

### 2.3. Other Factors That Contribute to Functional Mitral Regurgitation

The geometrical distortions of the mitral annulus caused by LV dilatation contribute to FMR [20,21]. The annulus’ geometry (shape and dimension) and motion (sphincter function) are impaired. The typical saddle morphology is altered so that the annulus tends to assume a monoplane flattened shape. In addition, the degree of this geometric distortion is greater in anterior myocardial infarction [22,23]. Coupled with the imbalance between tethering and closing forces, mitral valve tenting results in coaptation loss that aggravates FMR.

Another contributing factor that may worsen FMR in patients with HF is mitral annular enlargement driven by left atrial dilatation in the presence of atrial fibrillation [24]. This mechanism is also known as atriogenic leaflet tethering, which implies the stretching of the posterior mitral leaflet across the LV wall by left atrial dilatation, with subsequent displacement/tethering of mitral valve leaflets away from the PMs [25].

Lastly, different studies suggest an organic contribution to functional mitral regurgitation. It was shown that loading conditions in patients with HF result in structural and biochemical alterations in MV leaflets, characterized by a significant increase of glycosaminoglycans and collagen. Moreover, it was reported that these valves in HF have greater concentrations of cells and a lower concentration of water. In turn, the greater concentration of cells indicates an upregulation of mitogenic activity, resulting in increased matrix synthesis. The loss of water, which is a major component of the extracellular matrix and contributes to tissue viscoelasticity, is likely responsible for reduced viscous relaxation. Therefore, these compositional alterations observed could have functional consequences on the valves resulting in stiffer, less extensible, and less viscous [26,27,28,29,30,31,32].

## 3. Mechanical Dyssynchrony

LV mechanical dyssynchrony could be a potential contributing factor to FMR by several mechanisms. First, the mechanical dyssynchrony of LV segments increases the tethering force in dual modes: displacement of PMs with consequent lack of leaflets coaptation and uncoordinated regional mechanical activation due to dyssynchrony both contribute in distorted MV apparatus geometry [33]. Moreover, LV dyssynchrony generates a positive pressure gradient between the LV and the left atrium due to altered timing of atrioventricular relaxation and contraction cycles, leading to a diastolic FMR during incomplete mitral valve closure [34]. Finally, LV dyssynchrony decreases LV contractility and consequently closing forces, impairing mitral valve tenting [35].

## 4. The Rationale of CRT

According to guidelines for managing HF, optimal medical therapy should be the first step in managing all patients with FMR [36]. CRT should be evaluated according to ESC Guidelines, which recommend cardiac resynchronization in patients with HF who meet ECG and LV function criteria or regardless of NYHA class or QRS width in patients who have an indication for ventricular pacing for high degree AV block [37]. The early beneficial effect of CRT is the reduction of FMR due to improved, coordinated timing of mechanical activation of PMs insertion sites. In contrast, the long-term decrease in FMR is secondary to LV reverse remodeling [38]. Acutely, the increased contraction efficiency due to global LV resynchronization also gives an important contribution to FMR reduction. This is especially true when resynchronization does not just involve apical segments, but also basal and mid-LV portions for a global effect [39]. Indeed, in CRT patients receiving an LV lead with multiple selectable electrodes (a quadripolar LV lead), the simultaneous recruitment of a wider portion of LV myocardium is associated with an acute electromechanical improvement [40,41,42]. However, in ischemic patients, the benefits of CRT on FMR may be limited in those with significant scar burden because the activation site may not be able to contribute to LV contractility. Furthermore, the implanting electrophysiologist cannot always place the LV lead in the “best position” to enhance myocardial viability. Electrophysiologists are often conditioned by coronary sinus anatomy, availability of coronary sinus branches, high capture threshold of LV lead, and capture of the diaphragm muscle.

Many studies have shown that CRT reduces FMR by increasing closing forces and, more importantly reducing tethering forces through “local” synchronization. Indeed, CRT acutely determines resynchronization of the PMs, leading to a shortening of MR duration and later onset of MR [43,44].

Other potential contributions of CRT may derive from reshaping annular geometry and function and decreasing sphericity indices [45,46]. This was paralleled by more normalized anatomy of the MV apparatus and decreased MR severity.

CRT has also been associated with left atrial reverse remodeling and a decrease in the burden of atrial fibrillation, which is implicated in mitral annular dilatation in patients with HF [47].

Regarding the long-term effects of CRT on FMR, they are mainly represented by the increase of closing forces and the global LV inverse remodeling. These results have been achieved after 3–6 months of CRT on top of optimized medical treatment and are even more evident during longer follow up [48].

In conclusion, immediate FMR reduction after CRT implantation is the expression of balancing of tethering and closing forces and results from an acute reduction in volume overload. The immediate FMR reduction also favors reverse remodeling and has been demonstrated to be a robust prognostic determinant after CRT. In the long-term, FMR reduction is the expression of LV healing and reverse remodeling (Figure 1).

## 5. Effects of CRT on Clinical Outcomes

In a post hoc analysis of the Cardiac Resynchronization—Heart Failure trial (CARE-HF), patients assigned to CRT were more likely to have an improvement in MR and a reduction in NT-proBNP after three months of follow-up. This finding suggests that this factor may predict or mediate some of the long-term response to CRT. The baseline degree of MR was not reported [13].

Previous studies (MIRACLE, MUSTIC) reported at least the FMR grade before and after CRT implantation and demonstrated a small but significant reduction in FMR severity at 3–6 months. In the Multicenter InSync Randomized Clinical Evaluation (MIRACLE) trial, MR decreased significantly at three and six months in patients with moderate-to-severe HF who received CRT [49]. Linde C et al. analyzed the long-term results of MUSTIC trial. Regarding MR, they found that, in patients in sinus rhythm, MR had decreased markedly by 24% at six months, by 39% at nine months, and by 45% at twelve months. For the atrial fibrillation group, MR decreased by 35% at nine months and 50% by twelve months compared with baseline values [50]. From these data, it seems that there is no difference in FMR reduction between patients with atrial fibrillation and in sinus rhythm. Different results from another study reported a more common FMR improvement in CRT recipients in sinus rhythm vs. atrial fibrillation, despite a similar degree of LV remodeling [51].

It is important to underline that in these early 2000s trials, MR evaluation was limited by heterogeneity in the definition of its mechanism and the methods of quantification. Indeed, regarding the methods of MR quantification, MR jet area and MR jet area/LA area were often used. However, both these methods may be subject to errors: in many cases, the regurgitant jet is eccentric due to asymmetric tenting of the posterior leaflet, and consequently, the size and width of the color jet may be underestimated. Moreover, while the first studies and landmark trials on CRT were not specifically evaluated to address the reduction of FMR after biventricular pacing, and only qualitative MR assessment was described, the latest studies provided a quantitative and more detailed assessment of MR (Table 1). In a post hoc analysis of the MADIT CRT trial, MR was graded as “none” in 2%, “mild” in 83%, “moderate” in 13%, and “severe” in 2% of patients at baseline. At a 12-month follow-up, most patients had no change in MR grade (84% in the ICD-only group versus 82% in the CRT-D group), although a greater number of patients worsened in the ICD-only group and improved in the CRT-D group (*p* = 0.016). These results suggest that CRT at least stabilizes and does not worsen FMR.

The persistence of FMR is associated with worse outcomes, as demonstrated by an observational study by Cabrera Bueno et al. This study included 32 patients (42%) with baseline moderate or severe FMR. However, in most of the study population (65.7%), FMR remained unchanged or worsened [52].

**Table 1 cells-11-02407-t001:** Trials and studies evaluating prevalence of FMR of various degrees in CRT candidates.

First Author, Year [Ref]	Type of Study	Number of Patients	Method of FMR Quantification	Severe FMR (%)	Prevalent Degree in Study Population (% of Study Population)	Reduction of FMR Degree (% of Study Population)	No Change FMR Degree (% of Study Population)	Worsening FMR Degree (% of Study Population)	Main Results
Cabrera Bueno et al., 2009 [48]	Observational	76	EROA	42	Non severe 58	35	66	0	Higher rate of clinical events and major arrhythmic events, in severe group
Solomon et al., 2010 [47]	Randomized controlled	749 (CRT-D arm)	MR Jet area	2	Mild 83	15.3	81.9	2.8	CRT stabilizes and does not worsen FMR
Van Bommel et al., 2012 [53]	Prospective	98	VC, EROA	2	Moderate-severe 63	49	51	0	MR improvers had better survival
Di Biase et al., 2012 [54]	Multicentre Retrospective	794	MR Jet area, VC, EROA	35	Mild-moderate 51	45	43	12	Basline MR, change in MR at 3-month follow-up strongly associated with CRT response.
Verhaert et al., 2012 [50]	Retrospective observational	266	VC, EROA	5	Mild 31 Moderate 25	100	0	0	Larger MR decrease and smaller residual MR are predictors of a better outcome. MR improvement is seen more frequent in patients with adavanced MR at baseline.
Cipriani et al., 2016 [55]	Prospective obervational	916	Multiparametric	55	Moderate or more 55	74	0	26	Worse prognosis if MR persistance or worsening.

FMR: functional mitral regurgitation; CRT: cardiac resynchronization therapy; EROA: effective regurgitant orifice area; VC: vena contracta; MR: mitral regurgitation.

On the other hand, Verheart et al., in a retrospective observational study, reported that patients with significant FMR pre-CRT experienced more reverse remodeling, indicating that severe FMR does not interfere with the effects of CRT [53]. All patients experienced an early decrease in FMR after CRT implantation, followed by LV reverse remodeling at the six-month follow-up, but most of the patients (84%) had trivial or mild MR.

Patients with moderate to severe FMR have been so far underrepresented in clinical studies. Therefore, Van Bommel and colleagues exclusively included patients with grade three and four FMR (according to ACC/AHA guidelines) and high operative risk for MV surgery in a prospective study [54,55]. It was demonstrated that CRT reduces the severity of FMR at six-month follow-up in 49% (42 patients) of the study population. The novel finding in this study was that a reduction of ≥1 grade of MR 6 months after CRT (MR improvement) also resulted in superior survival during long-term follow-up. Specifically, the 1- and 2-year survival rates were 97% and 92% in MR improvers compared with 88% and 67% in MR non-improvers.

If FMR persistence in CRT patients is associated with a worse outcome, it is crucial to proceed to FMR correction in non-responders and, therefore, to establish the optimal timing for FMR correction. Regarding this issue, useful information was provided by a multicentre retrospective study by Di Biase L et al. MR severity and LV reverse remodeling at baseline and at three and twelve months after CRT implantation were assessed [56]. Advanced FMR was present in 35% of the total population. Improvement of FMR was relatively more frequent in patients with advanced MR at baseline (63% of all severe FMR patients) than in subjects with mild/moderate FMR (33% of all mild/moderate patients). Ischemic etiology was related to less improvement. Moreover, it was found that improvement in FMR at 3-month follow-up (>1 grade) predicted response to CRT. Therefore, the authors proposed to delay the decision for MV treatment to the three-month follow-up because, at this time point, some patients could have FMR improved by CRT alone, not requiring any further interventions.

In conclusion, data from clinical studies and trials suggest that it is possible to achieve MR improvement early after CRT implantation, but more commonly after three or six months and more often in sinus rhythm patients. Sometimes CRT neither modifies nor worsens MR, but this is associated with an unfavorable long-term survival.

## 6. What to Do in CRT Non-Responders with Significant Mitral Regurgitation?

In 20–25% of CRT patients, significant MR persists. These numbers concern patients with grade ≥2 FMR and are often found to be CRT non-responders. It is unclear why these patients do not benefit from CRT compared to their counterparts. In ischemic patients, significant scar burden may hamper the benefits of CRT on FMR. Different studies suggest that patients with baseline greater ventricular volumes and greater EROA, i.e., greater severity of FMR, experienced less reverse remodeling than patients without FMR at baseline. Therefore, globally available data support the idea that maximum benefit in CRT is obtained in patients ill enough to get improvement but not too ill to be able to get any benefit.

Catheter-based treatment with the MitraClip (Abbott Vascular Structural Heart, Menlo Park, California) and, more recently, PASCAL clip (Edwards Lifescience, Irvine, CA, USA) have been proposed as an additional therapeutic option in selected patients with FMR [57,58]. Due to their relatively low-risk profile compared with surgery, these devices provide the potential to fill the therapeutic gaps left in CRT non-responder patients.

So far, two randomized studies have been published focusing on subjects with FMR: the MITRA FR trial and the COAPT trial. In the French study MITRA FR, 307 patients with HF and severe FMR were randomly assigned to edge-to-edge mitral valve repair plus optimal medical treatment or optimal medical treatment alone [59]. Prior CRT implantation was present in 27% of the study population. MC successfully reduced MR to Grade 2 + or less in 92% of patients at the time of hospital discharge; however, no impact on the primary outcome of all-cause mortality or HF re-hospitalization at 1-year follow-up was demonstrated. The US trial COAPT included 615 patients randomly assigned to edge-to-edge MC repair plus optimal medical treatment or optimal medical treatment alone. Inclusion criteria were symptomatic HF and moderate-severe or severe FMR [60]. Prior CRT used was 36% of the patient population. The trial showed positive data in reducing FMR degree (95% of the total population) and in achieving a reduction in hospitalizations for HF and all-cause mortality within 24 months of follow-up.

Many reasons have been suggested to justify this discrepancy in outcomes between the trials. (Table 2)

The main difference was in patient selection: the COAPT trial included patients with moderate left ventricular dilatation and more severe MR. Conversely, the MITRA-FR trial included patients with extreme left ventricular dilatation and less severe FMR (EROA < 30 mm^2^), showing that these patients are unlikely to benefit from transcatheter mitral valve repair. This finding is in line with post hoc subgroup analysis of COAPT, demonstrating that patients with EROA ≤ 30 mm^2^ and left ventricular end-diastolic volume > 96 mL/m^2^ (10.2% of the COAPT study population) demonstrated no change in all-cause mortality or heart failure re-hospitalization one year after MC implantation. Therefore, the differences in outcomes between these two trials suggest that patient selection is critical for reaching procedural clinical success. Additionally, returning to resynchronization therapy, it can probably be said that even for CRT the key aspect is represented by patient selection. However, the identikit of the perfect candidate (responder) for CRT, is not yet well established.

In patients with end-stage HF, not responding to optimal medical therapy or to edge-to-edge mitral valve repair, left ventricular assist device (LVAD) therapy could be considered as a bridge to transplantation and/or destination therapy. It has been reported that mechanical unloading with LVAD leads to a reduction in the severity of FMR in these patients. However, in this case, moderate or severe FMR persists in nearly 20%–30% of patients. Risk factors for residual FMR include younger age, female gender, non-ischemic heart failure etiology, and pre-implant right ventricular dysfunction. Residual MR is associated with an increased risk of right heart failure and renal failure and a trend toward mortality on LVAD support.

## 7. Open Issues and Future Perspectives

### 7.1. Need to Standardization

Firstly, it is important to use the same language: there is a need to standardize the methodology to assess the severity of FMR. Being a dynamic lesion is a feature of FMR in patients with HF; accordingly, its severity may vary over time. FMR is also very sensitive to loading conditions, and many patients with HF use drugs that modify preload. Therefore, using the traditional color Doppler flow mapping in the left atrium and measuring the regurgitant jet width to quantify FMR may lead to errors. The quantitative method using the Doppler measurement of stroke volumes or, when feasible, the analysis of the flow convergence zone using the PISA method is more accurate. It has been shown that the regurgitant volume and, specifically, the EROA are less load-dependent and more reliable. In moderate MR, exercise echocardiography may be useful for quantifying exercise-induced changes in mitral regurgitation [37].

### 7.2. Medical Therapy

In this context, an important unresolved issue is understanding when patients are truly refractory to medical therapy. The severity of FMR should be reassessed after optimized medical treatmDent. The latest AHA/ACC valve guidelines suggest the supervision of a cardiologist expert in the treatment of patients with HF to achieve optimal results and to determine with the multidisciplinary team when symptoms are truly refractory to guideline-directed medical therapy before decisions are made for further treatments. In addition, it is important that medical therapy should be up-titrated to the maximum tolerated dose to achieve the greatest possible benefit.

### 7.3. Technical Issues

LV ejection fraction should be carefully addressed when significant FMR is concomitantly present. For this reason, proper LV function assessment plays a central role and may influence the timing for CRT therapy, which might affect the CRT response. Indeed, LV ejection fraction overestimation in patients with severe FMR in some cases might exclude patients from the CRT indication and, consequently, prolong the timing for intervention. Moreover, following the HF treatment flow chart as proposed by European guidelines, those patients treated with Sacubitril/Valsartan might improve their LV function, preventing CRT implantation. However, some of them might already have or develop dyssynchrony without being suitable candidates for CRT due to LVEF >35% under Sacubitril/Valsartan effects.

A second aspect concerns mechanical dyssynchrony in the absence of electrical dyssynchrony. Data showing LV dyssynchrony in patients with HF and narrow QRS or effectiveness of CRT in non-LBBB ECG suggest that further randomized studies are needed and, possibly, CRT indications updated. Indeed, echocardiography and its advanced tools (speckle tracking, 3D LV ejection fraction, and tissue Doppler imaging) allow the detection of mechanical dyssynchrony in those cases where electrical dyssynchrony is absent and, consequently, they might play a role in the future as adjunctive techniques for best patient selection and CRT optimization.

### 7.4. Defining the Right Time of Intervention

The maximum benefit of CRT on FMR and LV remodeling should be observed in those patients with at least mild MR, not reaching a severe degree, which may represent a too late and perhaps irreversible stage of LV dysfunction and dilatation to receive benefit from electrical therapy. However, dyssynchrony is only one component of a highly complex pathology, and consequently, CRT alone has some intrinsic limitations for FMR resolution. Technical improvements in the field of electrophysiology with new leads and algorithms are going to increase the rate of success of such therapy. All the available tools, from medical therapy to MR correction, should be, therefore, considered in order to correctly address FMR and reverse the vicious cycle of HF-FMR. Further investigations are warranted to elucidate the potential benefit of combining CRT with specific therapeutic approaches for the mitral valve in patients with FMR and indications of CRT, like surgical annuloplasty or percutaneous therapies.

## 8. Conclusions

FMR, in the context of HF, is not an innocent bystander but significantly impacts survival. Consequently, FMR and HF should not be treated as two different pathologies but should be considered a unique, complex syndrome. Therefore, a multilevel approach aiming at addressing all the underlying mechanisms and complications of HF should be preferred. In this perspective, the multidisciplinary Heart Team, with multiple subspecialities, will play a central role to provide the best approach to such a multifaceted and life-threatening condition.

## Figures and Tables

**Figure 1 cells-11-02407-f001:**
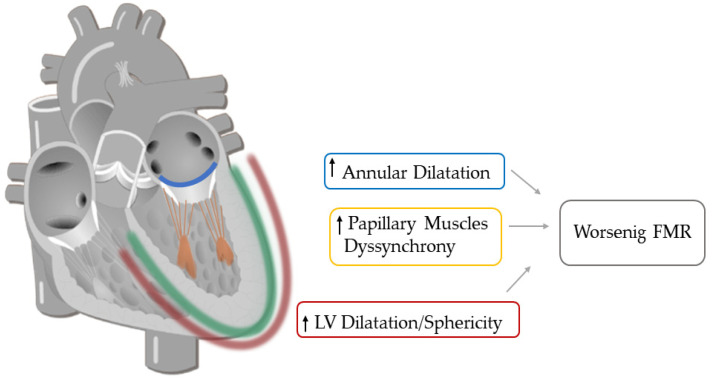
Exercise-induced dynamic geometric changes in the LV and mitral valve apparatus thus causing MR worsening.

**Table 2 cells-11-02407-t002:** Variables that Aid in Predicting and Monitoring MR Improvement.

Predictor Category	Predictors of MR Improvement	Reference
**Clinical parameters**	ΔQRS (at least 20 ms) after CRT	[61]
	QRS narrowing after CRT	[62]
	Older age	[63]
	Baseline longer QRS duration	[63]
**Echo imaging**	Combination of the presence of MR and viability in the region of the pacing	[39]
	baseline tenting area < 3.8 cm	[51]
	MR at baseline	[54]
	Change in MR at 3-month follow-up	[54]
	Increase of %10 LVEF	[55]
	Baseline tenting area	[62]
	Septal-lateral delay by TDI	[63]
	measurement of systolic dyssynchrony by TDI	[64]
	Time to- peak 2-DRS between inferior and anterior LV segments of >110 ms	[65]
	preserved radial strain in posterior segments assessed by 2-DRS	[65]
	MR jet area/left atrium area ratio<40%	[65]
	anteroseptal to posterior wall radial strain dyssynchrony >200 ms	[66]
	lack of severe left ventricular dilatation (end-systolic dimension index <29 mm/m^2^ )	[66]
	lack of echocardiographic scar at papillary muscle insertion sites	[66]
	**Predictors of Lack of MR Improvement**	
**Clinical parameters**	Chronic AF	[55]
**Echo/CT imaging**	MR at baseline	[54]
	Change in MR at 3-month follow-up	[54]
	Baseline moderate MR	[55]
	≥25% of LVWT <6 mm inclusive of at least one papillary muscle insertion using CT	[67]
**Biomarkers**	Higher levels of galectin 3	[68]

CRT: cardiac resynchronization therapy; TDI: tissue Doppler imaging; 2DRS:2D radial strain; MR: mitral regurgitation; LVWT: left ventricular wall thickness; CT: computed tomography; AF: atrial fibrillation.

## Data Availability

Not applicable.

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
