# Peer review of "The Role of Cardiac Resynchronization Therapy for the Management of Functional Mitral Regurgitation"

_cells, 2022, doi:10.3390/cells11152407_

Round 1

Reviewer 1 Report

The authors overviewed in this review article the detailed pathophysiological and clinical impact of cardiac resynchronization therapy on functional mitral regurgitation. Heart failure and functional mitral regurgitation have a deep association, and cardiac resynchronization therapy should affect both of them. The topic is of great concern for many readers and the description is easy to follow.

1. There are several grammatical errors that should be corrected.

2. Could the authors explain the reason why functional mitral regurgitation does not improve against cardiac resynchronization therapy in some patients?

 3. How about the patients with too advanced heart failure and functional mitral regurgitation? Functional mitral regurgitation often improves following implantation of durable left ventricular assist devices. 

Reviewer 2 Report

This review points out the effects and problems of CRT for functional mitral regurgitation (FMR) and describes them in detail. We would like to ask for corrections and additions to the following comments.

#1 It is clear that a good indication for CRT is a prolonged QRS, including CLBBB, there are many cases in which the QRS is not necessarily prolonged, even with FMR. Although this article focuses on CRT for FMR, it is necessary to reiterate the position of CRT in treating FMR, including the guidelines.

#2 The entire article is quite long. Since the title is FMR and CRT, please focus on that and make the rest more concise.

#3 In figure 1, the coloring makes sense, but the color at "annual dilation" is difficult to understand. In addition, please explain the abbreviation.

Round 2

Reviewer 1 Report

There are no further comments.